# Effectiveness of menstrual hygiene management training to enhance knowledge, attitude, and practice among adolescents in Sindhupalchowk, Nepal

Swastika Shrestha[1]*, Saki Thapa[1], Bikram Bucha[2], Safal Kunwar[1], Bigyan Subedi[1], Aishwarya Rani Singh[3], Durga Datta Chapagain[4], Raghu Dhital[1], Maxine Caws[1]

1 Birat Nepal Medical Trust (BNMT), Lazimpat, Kathmandu, Nepal, 2 Karolinska Institute, Stockholm, Sweden, 3 Putali Nepal, Kathmandu, Nepal, 4 Yonsei University, Seoul, South Korea

* swastika@bnmt.org.np

**Data Availability Statement:** Data cannot be shared publicly because consent was not obtained

## Abstract

### Background

Menstrual Health (MH) knowledge, attitude and practice (KAP) are highly affected by access to information regarding menstruation. Despite being included in the school curriculum, Sexual and Reproductive Health (SRH) education is often not delivered in practice. School-based educational interventions have been shown to be effective in promoting MH.

### Methods

A school-based study was conducted in Indrawati rural municipality of Sindhupalchowk district in Nepal. 427 participants (175 boys and 252 girls), aged 11–13, completed a questionnaire evaluating MH KAP before receiving a structured training module on MH provided by experienced trainers from Putali Nepal using the Menstrupedia tool. The questionnaire was repeated one month after the training. Pre and post intervention scores were compared to determine the effect of the intervention. Focus group discussions were also conducted to understand the perceptions of participants toward SRH teaching. Association of independent socio-demographic with dependent variables knowledge and attitude towards menstrual health were analyzed using MANOVA test. The Wilcoxon signed-rank test was used to compare the median outcome of the pre and post-test attitude and knowledge. The maximum possible score was 6 for MH knowledge. The total attitude score ranged 14 to 70.

### Results

The median knowledge score increased by 1 point (p = <0.001) and the median attitude score by 5 points (p = <0.001), one month after delivery of the intervention. Higher knowledge scores were significantly associated with Hindu religion, female gender, higher father's literacy, and mothers in an informal occupation on multivariate analysis. Higher attitude scores were significantly associated with Hindu religion while lower attitude scores were associated with a mother in an informal occupation.

for public data sharing of full transcripts during the informed consent process, due to the sensitive nature of the questions. Data is securely stored in controlled access secure data storage at Birat Nepal Medical Trust (BNMT). Data is available from the BNMT data access Committee (contact: Kritika Dixit, research manager BNMT; kritika@bnmt.org. np) for researchers who meet the criteria for access to confidential data.

**Funding:** The research was funded by a personal funding made to the Birat Nepal Medical Trust by Mr Frank Guthrie. The funders had no role in study design, data collection and analysis, decision to publish, or preparation of the manuscript

**Competing interests:** The authors have declared that no competing interests exist.

## Conclusion

The Menstrupedia comic educational intervention improved knowledge and attitude towards menstruation among Nepali adolescents. A scale-up of the Menstrupedia based intervention would significantly change knowledge and attitude towards menstruation in Nepali adolescents.

## Introduction

Menstrual health (MH) is an indispensable component of physical health and overall wellbeing in women. Although a normal physiological process, many cultures around the world view menstruation as a subject of taboo and have imposed various degrees of limitations on menstruating women [1–5]. Due to the stigma surrounding menstruation, adolescent girls in developing countries are often uninformed about, and thus are unprepared for, menarche. This leads to misconceptions and unhygienic practices during menstruation [6]. Attitude towards menstruation is affected by the social norms, expectations, and beliefs about how women should feel, act, and behave during the menstrual cycle [7]. This inevitably affects the MH of women and girls- which is defined as a state of complete physical, mental, and social well-being and not merely the absence of disease or infirmity, in relation to the menstrual cycle [8].

Knowledge, attitude, and practice (KAP) regarding menstruation are interlinked and are highly affected by access to information regarding menstruation and MH [9–11]. Although the school curriculum in Nepal includes sexual and reproductive health (SRH) education, it is often not implemented, largely because teachers themselves are reluctant to discuss sensitive topics surrounding SRH [12, 13]. In such contexts educational interventions including both the students and teachers can be effective in promoting MH [14–16].

MH education should not only include females. Men and boys play an important role in either supporting or preventing women and girls in managing menstruation effectively across different social arenas including, but not limited to, household, community, school and work, as boyfriends, husbands, fathers, brothers, students, peers, and teachers [17]. Thus, the intervention applied here included all adolescents, irrespective of their gender identities. The research evaluated the effectiveness of a structured training module using the Menstrupedia comic (www.menstrupedia.com). Menstrupedia is an adolescent friendly guide to periods which facilitates understanding of the dynamics of the menstrual cycle and the pubescent changes that occur in the body, thereby encouraging girls to take appropriate actions to manage their menstrual cycle and stay healthy and active whenever possible during their periods. We evaluated and compared the KAP regarding MH among school-going adolescents in Indrawati municipality of Sindapulchowk before and after the Menstrupedia training.

Underpinning Sustainable Development Goals (SDG) [18], MH is a right for every woman and girl. Imparting MH education is an essential step towards ensuring the right to MH for women and girls. Although a part of the school curriculum, schools in Nepal have not been able to deliver the SRH education effectively. In such context, development and evaluation of evidence-based interventions is crucial for supporting political commitment and investment in improving school-based teaching for MH and the broader comprehensive sexuality education (CSE) curriculum. Despite the severe and deeply rooted stigma surrounding menstruation in Nepal, which has multidimensional negative impacts on the lifelong wellbeing of women and girls, there is a paucity of published studies evaluating MH educational interventions in

the Nepalese context. Therefore, we designed the present study to evaluate the effectiveness of a school-based menstrual education program in improving menstrual knowledge and attitude among adolescents in Nepal.

## Materials and methods

### Setting

This is a pre-post longitudinal school-based study conducted in Indrawati rural municipality of Sindhupalchowk district, Nepal where there is a high prevalence of underage marriage [19]. The district was also badly affected by the 2015 earthquake in Nepal, following which there were increases in the lack of segregated toilets, poor living conditions and lack of privacy for women and girls affecting their SRH [20].

### Intervention

The intervention consisted of a structured training module using the Menstrupedia comic, which was taught to the students by experienced trainers from Putali Nepal (https://putali-nepal.com/). Putali Nepal is a non-governmental organization experienced in educational interventions to improve menstrual hygiene knowledge among adolescents and combat prevailing stigmas. The adolescents were provided with information on MH and menstrual hygiene management (MHM) using a sixty-minute animation video which incorporated characters featured in the Menstrupedia comic. The training sessions were conducted for each grade in their normal classrooms with students of all genders present. Each session consisted of no more than 30 students. The contents of the video introduced the adolescents to physical changes during puberty, healthy diet, menstruation, things to pay attention to during the first menstrual period, techniques to reduce menstrual cramping, the physiology of the menstrual cycle, premenstrual syndrome, use and management of sanitary pads. The students were encouraged to ask questions and participate in the games which were part of the training.

### Study participants and sampling process

There are 12 government schools in Indrawati municipality, Sindhupalchowk. Assuming the baseline knowledge score of menstrual hygiene for students to be 50%, 385 students would be sufficient to estimate the true proportion in the study population with 95% confidence. However, due to COVID disruption to school attendance, we were unable to control the sample size of students completing both pre and post intervention questionnaires, and therefore the final matched pair analysis was performed on collected data without a predetermined power calculation.

The study was conducted in 9 of the 12 public schools in Indrawati municipality, Sindhupalchowk. The three excluded schools already had an alternative active intervention for MH and advocacy under the BNMT Amplify Change project. The study population included adolescent students aged 11–13 (classes 5, 6 and 7) attending the participating government schools in the municipality.

### Ethics statement

Ethical approval for the study was obtained from the Nepal Health Research Council (NHRC) [Ethical review board (ERB) Protocol Registration number: 356/2021]. Approval was also obtained from the principals of the participating schools. The Menstrupedia classes were conducted according to a schedule agreed with the school authorities. The students from selected classes were verbally given information about the study and the confidentiality of their

personal information, and then provided with an opportunity to ask questions. They were provided with a participant information sheet containing information regarding the study which included the contact number of the researcher so that the participants or parents/guardians could inquire if they had any questions regarding the study. Written informed consent was obtained from parents or guardians. Students also received an assent form to sign if they were willing to participate in the study. The students were asked to discuss the study with their parents and return the signed consent and assent forms.

## Data collection instruments and procedures

The data collection was conducted between December 2021 and February 2022. A self-administered, structured, close-ended, and anonymous questionnaire consisting of questions on socio-demographic details, experience, knowledge, attitude, and practice regarding menstruation and menstrual hygiene was used as the study tool for the quantitative aspect of the study. The questionnaire was adapted to the Nepali context from published studies and open access tools [11, 21–25]. The students were given as much time as required to complete the questionnaire under the supervision of the investigator. All students had normally completed within 45 minutes. Following the data collection, the students participated in the Menstrupedia comic-based teaching conducted by experienced facilitators from Putali Nepal in partnership with the class teachers. Questions from the participants relating to menstrual and reproductive health were discussed during the training. One month following the single training session, the same KAP questionnaire was completed by the students to gather the post intervention data on KAP.

The qualitative evaluation of the intervention consisted of six focus group discussions (FGDs), 3 with girls and 3 with boys, with selected students in the schools. Each FGD included either 10 boys or 10 girls. Among the students who were interested in participating in the FGDs and whose parents had consented to their participation, 60 were chosen randomly to be included in the FGD. The FGD included topics pertaining to the barriers to receiving effective SRH education/information in the school, preferences regarding delivery of SRH teaching and feedback regarding the Menstrupedia training session. The FGDs were conducted in the Nepali language and were recorded. Each FGD lasted approximately 40 min and was transcribed verbatim. The transcript was then translated into English by the researcher who took the interviews and was validated by the co-investigators. The corresponding author identified the codes and generated the themes which were validated by the project manager who is also one of the co-authors. The transcribed document was manually analyzed by reading and re-reading the document line by line to generate codes. An inductive approach was used to derive codes from the content of the transcripts. Significant statements were identified. The meanings of each significant statement were formulated into codes, based on related words or phrases that were mentioned by the interviewees. New codes were added as they emerged during the coding process. Coding was continued until no new codes were identified. After reviewing the codes, a list of themes was generated [26]. The document was shared online amongst the researchers to facilitate analysis and allow transparency between the coding and analysis work. Finally, the codes were discussed amongst the researchers until agreement was reached on the development of themes. Four major themes were identified: teaching style in the schools, use of multimedia resources for teaching, impact on MHM practice and attitude and co-education during the training.

## Data analysis

There were seven questions related to the demographic variables, six multiple choice questions regarding knowledge, 35 questions regarding practice and fourteen statements regarding attitude on menstruation. Each correct response to knowledge questions was scored with one

point, whereas any incorrect or 'do not know' response scored 0. The total knowledge score for each individual was calculated by adding the scores for each correct answer. Detailed practice questions (n = 35) were completed only by the girls who had reached menarche.

Attitude was evaluated using responses to 3 positive and 11 negative statements on Likert's five-point scale. (strongly agree, agree, neutral, disagree or strongly disagree). The score of positive statement ranged from five to one (strongly agree-5, agree-4, neutral-3, disagree-2, strongly disagree-1 and do not know/cannot say- 0) and score for negative statement ranged from one to five (strongly agree-1, agree-2, neutral-3, disagree-4, strongly disagree-5 and don't know/cannot say- 0). The total attitude score for each individual was calculated by adding the scores for each statement.

Pre-intervention test data was collected from a total of 580 participants, which included 266 boys and 314 girls. The large difference in individuals attending school in the pre and post intervention groups was caused by the COVID disruptions and significantly reduced our matched sample for analysis. There was a 19% (n = 111) loss to follow up in the post intervention data collection due to an upsurge in COVID cases which led to fewer students attending the school during the post intervention data collection time. Post-intervention test data was collected from a total of 469 participants, which included 201 boys and 268 girls. Among the participants 427 students (175 boys and 252 girls) participated in both the pre and post intervention tests. Pre-test data of the participants was used to analyze the menstrual hygiene practice as well as associations between the dependent variables (knowledge and attitude) with the independent variables (sociodemographic variables). Knowledge and attitude scores of only the participants who took both the pre and post tests were compared.

The data was entered using ODK data collection mobile software [27] by junior researchers: Sujata Adhikari and Bikram Bucha. Entries were checked for accuracy and completeness by Dr Swastika Shrestha. Data analysis was conducted on STATA version 13 [28]. Association of independent socio-demographic variables ethnicity, religion, parent's education, parent's occupation with dependent variables knowledge and attitude towards menstrual hygiene were analyzed using MANOVA test. A (probability value) P value of $\leq 0.05$ was considered to indicate a statistically significant association. The Wilcoxon signed-rank test was used to compare the median outcome of the pre and posttest attitude and knowledge.

## Results

### The sociodemographic variables

Sociodemographic variables of the participants in the pre-test group, post-test group and the matched analysis group are shown in Table 1.

### MH knowledge

The maximum possible score was 6 for MH knowledge. A multivariate MANOVA test statistics was conducted for each predictor variable with knowledge (Table 2). The multivariate result was significant for Religion, F = 7.49, p = 0.0006; Mother's occupation, F = 4.16, p = 0.01; Father's literacy, F = 2.66, p = .04; and Gender F = 4.6, p = 0.03. For the pre-menstrupedia teaching group, there was a significant difference in menstrual knowledge score between the three religious groups: Hindu (n = 281), Buddhist (n = 132) or 'other religion' (n = 14). The median score for Hindus was 3, for buddhists 2, and for 'other religious affiliation' the median score was 2. There was no significant correlation between socioecomic status indicators (literacy level and educational status of parents) and religion among the participants, suggesting that this was not due to socioeconomic differences between the religious groups.

**Table 1. Sociodemographic variables of student sample.**

| Sociodemographic variables | | All pre-test participants | All post-test participants | Matched group |
|---|---|---|---|---|
| | | Frequency (%) | Frequency (%) | Frequency (%) |
| | | (N = 580) | (N = 469) | (N = 427) |
| Religion | Hindu | 400 (68.9%) | 304 (64.8%) | 281 (65.8%) |
| | Buddhist | 158 (27.2%) | 139 (29.6%) | 132 (30.9%) |
| | Others | 22 (3.7%) | 26 (5.5%) | 14 (3.2%) |
| Father's literacy level | Illiterate | 61 (10.5%) | 55 (11.7%) | 41 (9.6%) |
| | Basic literacy | 233 (40.1%) | 193 (41.1%) | 171 (40.1%) |
| | Primary school | 197 (33.9%) | 136 (29.0%) | 142 (33.3%) |
| | Secondary school or above | 83 (14.3%) | 72 (15.3%) | 69 (16.6%) |
| | No response | 6 (1.0%) | 13 (2.7%) | 4 (0.9%) |
| Mother's literacy level | Illiterate | 130 (22.4%) | 126 (26.8%) | 85 (19.9%) |
| | Basic literacy | 256 (44.1%) | 204 (43.5%) | 188 (44.0%) |
| | Primary school | 114 (19.7%) | 82 (17.4%) | 90 (21.1%) |
| | Secondary school or above | 67 (11.5%) | 48 (10.2%) | 55 (12.8%) |
| | No response | 13 (2.2%) | 9 (1.9%) | 9 (2.2%) |
| Father's occupation | Unemployed | 27 (4.6%) | 15 (3.2%) | 20 (4.6%) |
| | Formal employment | 97 (16.7%) | 76 (16.2%) | 74 (17.3%) |
| | Informal employment[1] | 456 (78.6%) | 378 (80.6%) | 333 (77.9%) |
| Mother's occupation | Unemployed | 162 (27.9%) | 138 (29.4%) | 118 (27.6%) |
| | Formal employment | 52 (8.7%) | 50 (10.6%) | 41 (9.6%) |
| | Informal employment[1] | 366 (63.1%) | 281 (59.9%) | 268 (2.7%) |

[1] Informal labour in the districts is mainly manual labour for men (eg. Construction work, sand mining) and carpet weaving, tailor, or stone sculpture for women.

Participants with a higher literacy level of fathers showed higher levels of menstrual knowledge. The median score of those whose fathers were 'illiterate', had 'basic literacy' or 'primary level education' was 2, while those whose father's had 'secondary level education or higher' was 3. Girls and boys also had statistically significant difference in menstrual knowledge with the median knowledge score for boys being 2 and of the girls being 2.5. Employment status of the mothers also showed a statistically significant association with levels of menstrual knowledge. The median knowledge score of those whose mothers were 'unemployed' was 2, for those in 'formal employement' was 1.5, and those with 'informal employment' was 3.

Analysing only the matched group with both pre and post test intervention scores (N = 427), the post intervention knowledge score was statistically significantly higher than the pre-intervention knowledge score (Wilcoxon signed-rank test; z = -5.23, p = <0.001). The median knowledge score increased by 1 point in the post test. The median pre test score was 2, while the median post test score was 3. Among the 6 questions asked, correct responses did not increase for two of the questions, relating to the source of menstrual blood and the normal duration of menstruation ne (Fig 1).

## MH attitude

The total attitude score (possible range = 14 to 70) was calculated by adding the responses to 14 Likert-scale (range = 1–5) attitude questions. A higher score represented a more positive attitude regarding menstruation. For the pre-test group, multivariate MANOVA test statistics showed that there was significant association of attitude with Religion, F = 5.93, p = 0.002; and Mother's occupation, F = 3.35, p = 0.03 (Table 3). There was a significant difference in

**Table 2. Association of independent variables with menstrual knowledge for matched pretest group.**

| Demographic variable | | N (%) | Median score | MANOVA test Wilks' lambda (P-value) |
|---|---|---|---|---|
| Religion | | | | |
| | Hindu | 281 (65.8%) | 3 | 0.0006* |
| | Buddhist | 132 (30.9%) | 2 | |
| | Other | 14 (3.2%) | 2 | |
| Gender | | | | |
| | Male | 175 | 2 | 0.03* |
| | Female | 252 | 3 | |
| Father's literacy | | | | |
| | Illiterate | 41 (9.6%) | 2 | 0.04* |
| | Basic | 171 (40.4%) | 2 | |
| | Primary | 142 (33.5%) | 2 | |
| | Secondary+ | 69 (16.3%) | 3 | |
| Mother's literacy | | | | |
| | Illiterate | 85 (20.3%) | 3 | 0.07 |
| | Basic | 188 (44.9%) | 2 | |
| | Primary | 90 (21.5%) | 2 | |
| | Secondary+ | 55 (13.2%) | 3 | |
| Father's occupation | Unemployed | 20 (4.6%) | 3 | 0.37 |
| | Formal employment | 74 (17.3%) | 2 | |
| | Informal employment[1] | 333 (77.9%) | 2 | |
| Mother's occupation | Unemployed | 118 (27.6%) | 2 | 0.01* |
| | Formal employment | 41 (9.6%) | 1 | |
| | Informal employment [1] | 268 (62.7%) | 3 | |

[1.] Informal labour in the districts is mainly manual labour for men (eg. Construction work, sand mining) and carpet weaving, tailor or stone sculpture for women.

[2.] * Indicates a significant P-value (≤0.05)

menstrual knowledge between the three religious groups: Hindu (n = 281), Bhuddist (n = 132) or other religion (n = 14). The median attitude score for the Hindu group was 35.5 (IQR = 29;43), for Bhuddists 31 (IQR = 25;39) and for 'other religious affiliation' the median score was 33.5 (IQR = 27;42). Employment status of the mothers also showed a significant association. The median attitude score of those whose mothers were unemployed was 36 (IQR = 30;44), with formal employement was 36 (IQR = 30;43), and those with informal employment was 34 (IQR = 27;40).

Analysing, only the matched group with both pre and post test intervention attitude scores (n = 427), the post intervention attitude score was statistically significantly higher than the pre-intervention attitude score (Wilcoxon signed rank test; z = -6.1, p = <0.001). The median attitude score increased by 5 points in the post test. The median pre test attitude score was 34, while the median post test attitude score was 39.

## Menstrual hygiene practice

Responses to questions about MH practice are shown in Tables 4 and 5. Ninety percent of the girls (n = 58/64; 90.6%) reported being aware about menstruation before attaining their menarche. The majority of female participants cited (n = 44/64; 75.8%) mothers as a source of information, followed by friends (n = 35/64; 60.3%), teachers (n = 16/64; 27.6%), books (n = 15/64; 25.8%), media (n = 10/64; 17.2%) and others (n = 4/64; 6.9%). Disposable sanitary

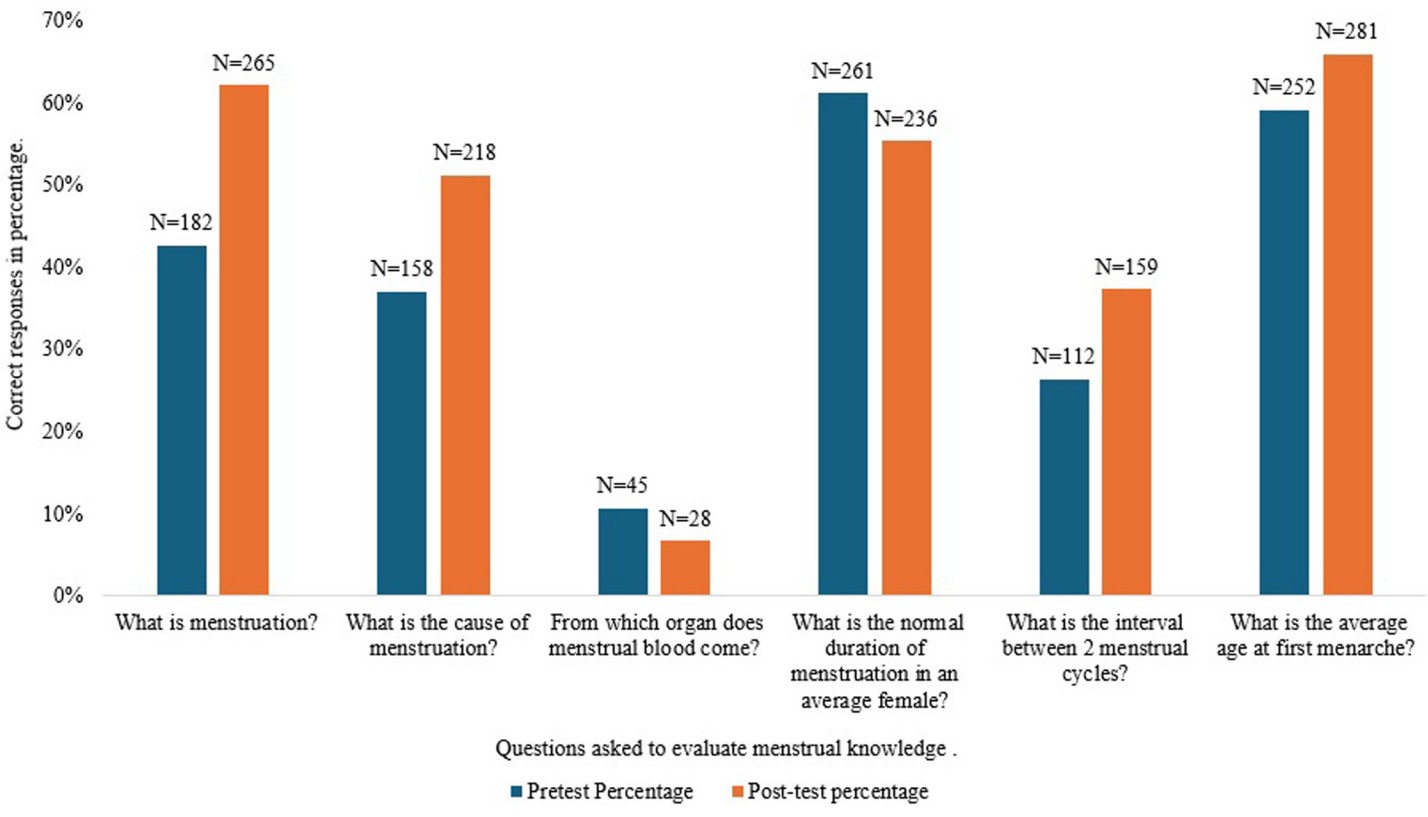

**Fig 1. The proportion of 427 students who gave correct answers to specific questions regarding menstrual knowledge before and after the Menstrupedia intervention.**

pads were the material used during menstruation both at home (n = 44/64; 68.7%) (n = 50/74; 67.5%) and at school (n = 49/64; 76.5%). Other most commonly used material at home was cloths (n = 14/64; 21.8%) while at school it was reusable sanitary pads (n = 15/64; 23.4%). Over half of the girls (n = 37/64; 57.8%) reported that their schools had the provision of only water and no soap available for handwashing. Less than one third (n = 18/64; 28.1%) reported having both soap and water in their schools. 1.5% (n = 1/64) reported having neither soap nor water for handwashing in their school. Regarding disposal of their menstrual materials while at home one third (n = 23/64; 35.9%) burnt the materials, and one third disposed of them in the household rubbish bin in the latrine (n = 20/64; 31.2%) while less common methods were disposing of sanitary materials by burying (n = 6/64; 9.3%) or throwing them into the latrine (n = 4/64; 6.6%)

While at school the majority of the girls disposed of their menstrual materials in the toilet bins (n = 36/64; 56.2%) A little less than a fifth of the girls (n = 12/64; 18.7%) reported transporting the materials back home for disposal while 15.6% (n = 10/64) reported disposing of materials into the latrine. The majority of the girls (n = 12/17; 70.5%) who used cloths as their menstrual material described drying them outside hanging in the sun after washing, 23.5% (n = 4/17) dried the materials outside but hiding the materials from public view while 5.8% (n = 1/17) dried the cloths inside and hidden. 52.9% (n = 9/17) of the girls did not cover the cloth with other materials while drying while 47.1% (n = 8/17) covered the cloths when drying. The most common restriction followed during menstruation was not attending the temple or participating in religious activities (n = 52/64; 81.2%) followed by not being allowed to touch

**Table 3. Association of independent variables with menstrual attitude for the matched pretest group.**

| Demographic variable | | N (%) | Median score | MAVOVA test Wilk's lambda (P-value) |
|---|---|---|---|---|
| Religion | | | | |
| | Hindu | 281 (65.8%) | 36 | 0.0002* |
| | Buddhist | 132 (30.9%) | 32 | |
| | Other | 14 (3.2%) | 33.5 | |
| Gender | | | | |
| | Male | 175 (40.9%) | 37 | 0.15 |
| | Female | 252 (59.1%) | 34 | |
| Father's literacy | | | | |
| | Illiterate | 41 (9.6%) | 36 | 0.3 |
| | Basic | 171 (40.4%) | 33 | |
| | Primary | 142 (33.5%) | 35 | |
| | Secondary+ | 69 (16.3%) | 34 | |
| Mother's literacy | | | | |
| | Illiterate | 85 (20.3%) | 33 | 0.98 |
| | Basic | 188 (44.9%) | 34 | |
| | Primary | 90 (21.5%) | 34.5 | |
| | Secondary+ | 55 (13.2%) | 35 | |
| Father's occupation[1] | Unemployed | 20 (4.6%) | 38 | 0.53 |
| | Formal employment | 74 (17.3%) | 36 | |
| | Informal employment | 333 (77.9%) | 34 | |
| Mother's occupation[1] | Unemployed | 118 (27.6%) | 36 | 0.03* |
| | Formal employment | 41 (9.6%) | 37 | |
| | Informal employment | 268 (62.7%) | 34 | |

[1.] Informal labour in the districts is mainly manual labour for men (eg. Construction work, sand mining) and carpet weaving, tailor or stone sculpture for women.

* Indicates a significant P-value ($\leq$0.05)

plants (n = 31/64; 48.4%), not being allowed to touch/sit with a male member of their family (n = 20/64; 31.2%), not being allowed to enter the kitchen (n = 16/64; 25.0%) and not being

**Table 4. Menstrual hygiene practice of female post-intervention test participants who had reached menarche.**

| Menstrual hygiene practice | N (%) | Total N |
|---|---|---|
| Knew about menstruation before attaining menarche: | | |
| 1. Yes | 58 (90.6%) | 64[1] |
| 2. No | 6 (9.3%) | |
| 3. No response | 0 | |
| Source of information: | | |
| 1. Mother | 44 (75.8%) | |
| 2. Teacher | 16 (27.6%) | |
| 3. Friends | 35 (60.3%) | |
| 4. Books | 15 (25.8%) | |
| 5. Media | 10 (17.2%) | 58 [2] |
| 6. Others | 4 (6.9%) | |

[1] The total number of girls who had attained menarche.

[2] The total number of girls who knew about menstruation before attaining menarche.

**Table 5. Practice of the girls pre and post intervention.**

| Girls' Menstrual hygiene practices | Pretest practice N (%) | Pretest practice Total N | Post test practice N (%) | Post test practice Total N |
|---|---|---|---|---|
| Materials used to absorb blood during menstruation (at home): | | | | |
| 1. Cloth/towel | 22 (34.4%) | | 14 (21.8%) | |
| 2. Disposable sanitary pad | 37(57.8%) | | 44 (68.7%) | |
| 3. Reusable sanitary pad | 2 (3.1%) | 64[1] | 7 (10.9%) | 64[1] |
| 4. Toilet paper | 0 | | 2 (3.1%) | |
| 5. Cotton wool | 0 | | 1 (1.5%) | |
| 6. Others | 1 (1.6%) | | 1 (1.5%) | |
| Materials used to absorb blood during menstruation (at school): | | | | |
| 1. Cloth/towel | 11 (17.2%) | | 5 (7.8%) | |
| 2. Disposable sanitary pad | 42 (65.6%) | | 49 (76.5%) | |
| 3. Reusable sanitary pad | 5 (7.8%) | 64 | 15 (23.4%) | 64 |
| 4. Toilet paper | 1 (1.5%) | | | |
| 5. Cotton wool | 0 | | | |
| 6. Others | 3 (4.7%) | | 0 | |
| Disposal of used disposable menstrual material after use at home: | | | | |
| 1. Into the latrine/toilet | 6 (9.4%) | | 4 (6.6%) | |
| 2. Burned | 15(23.4%) | | 23(35.9%) | |
| 3. Household rubbish (bin in latrine) | 11(17.2%) | 64 | 20 (31.2%) | 64 |
| 4. Household rubbish (bin not in the latrine) | 2(3.1%) | | 5 (7.8%) | |
| 5. Taken to community rubbish | 0 0 | | 4 (6.2%) | |
| 6. Bush/waterway/buried the material | 17(26.6%) | | 6 (9.3%) | |
| 7. Did not dispose of any materials | 9(14.1%) | | 1 (1.5%) | |
| 8. Other | 4(6.3%) | | 1 (1.5%) | |
| Disposal of used disposable menstrual material after use at school: | | | | |
| 1. Transported home to dispose | 16 (25%) | | 12 (18.7%) | |
| 2. Into the latrine/toilet | 9(14.1%) | | 10 (15.6%) | |
| 3. Bin in the latrine/toilet | 30(46.9%) | | 36 (56.2%) | |
| 4. Bin onsite but outside of the latrine/toilet | 1(1.5%) | 64 | 1 (1.5%) | 64 |
| 5. Community rubbish (not onsite) | 0 0 | | 1(1.5%) | |
| 6. Bush/waterway/buried the material | 1(1.5%) | | 2 (3.1%) | |
| 7. Burned the material | 0 0 | | 0 0 | |
| 8. Other | 7(10.9%) | | 2 (3.1%) | |
| Provision of both soap and water available at the handwashing facilities in school: | | | | |
| 1. Yes, water and soap | 20 (31.2%) | | 18 (28.1%) | |
| 2. Water only | 35 (54.6%) | 64 | 37 (57.8%) | 64 |
| 3. Soap only | 1 (1.5%) | | 8(12.5%) | |
| 4. Neither water nor soap | 8(12.5%) | | 1(1.5%) | |
| Washed genitals using soap: | | | | |
| 1. Never | 16 (25.0%) | | 44 (68.7%) | |
| 2. Sometimes | 19 (29.6%) | | 6 (9.3%) | |
| 3. Every time | 29 (45.3%) | 64 | 13 (20.3%) | 64 |
| 4. No response | 0 | | 1 (1.5%) | |
| Drying of the washed menstrual material: | | | | |
| 1. Outside (hanging in sun) | 16(57.1%) | | 12 (70.5%) | |

*(Continued)*

**Table 5.** (Continued)

|  | Pretest practice | | Post test practice | |
|---|---|---|---|---|
| **Girls' Menstrual hygiene practices** | **N (%)** | **Total N** | **N (%)** | **Total N** |
| 2. Outside (hidden) | 10(35.7%) | 28[2] | 4 (23.5%) | 17[2] |
| 3. Inside (hidden) | 1(3.6%) | | 1 (5.8%) | |
| 4. Other | 1(3.6%) | | 0 0 | |
| Cover the menstrual material with another cloth when drying: | | | | |
| 1. Yes | 11(39.3%) | | 8 (47.1%) | |
| 2. No | 16(57.1%) | 28 | 9 (52.9%) | 17 |
| 3. No response | 1(3.5%) | | 0 | |
| Restrictions followed in the household during menstruation: | | | | |
| 1. Not allowed to go to temple/participate in religious activities. | 48(75%) | | 52 (81.2%) | |
| 2. Not allowed to touch/sit with male member of family | 11(17.2%) | | 20 (31.2%) | |
| 3. Not allowed to enter the kitchen | 17(26.5%) | 64 | 16 (25.0%) | 64 |
| 4. Not allowed to touch plants | 31(48.4%) | | 31(48.4%) | |
| 5. Not allowed to touch books | 3(4.7%) | | 5(7.8%) | |
| 6. None of the above | 5(7.8%) | | 4(6.2%) | |
| 7. Others | 2(3.1%) | | 0 | |

[1] The total number of girls who had attained menarche.

[2] The total number of girls who used cloth pads as absorbent materials during their periods.

allowed to touch books (n = 5/64; 7.8%). Only 6.2% (n = 4/64) of the girls reported not having to follow any of the restrictions when they were menstruating. The pre and post intervention tests revealed that there were improved practices around managing menstrual cloths, however reporting of restrictive taboos increased.

Focus group discussions:

Four major themes were identified: teaching style in the schools, use of multimedia resources for teaching, impact on MHM practice and attitude and co-education during the training.

**Teaching style in schools.** SRH teaching practice in schools was explored during the FGDs. The students expressed discontent regarding the teaching methods surrounding SRH courses. They described teachers often having a one-way communication style of teaching, where the students were not encouraged to ask questions. Also, they described teachers using only verbal communication without visual aids to explain concepts. Students reflected that this approach made learning about organs that they could not see difficult.

The students shared that their teachers were hesitant to describe in detail about reproductive health and often laughed themselves during the lessons when teaching. The male students indicated that they would prefer a student friendly teaching environment where incorrect answers were not punished. Students stated that if teachers were not shy to talk about SRH issues then the students would have greater confidence talking about these issues. The students expressed that they would like it if the teachers incorporated learning games into the lessons.

*"We want someone who would teach us more clearly. Our sir doesn't make us understand the contents in the book as clearly as the trainers did." (12 Yo female)*

*"It could be easier for us to ask about reproductive health with a female teacher and we won't feel scared to ask." (11yo female)*

**Use of multimedia resource for teaching.** The students found the training session based on the Menstrupedia comic entertaining and informative. The use of multimedia during the teaching session was novel for the students, which they found interesting. Participants appreciated that the trainer stopped the video at times and asked questions, repeated the content, gave the students a chance to answer or pose questions on the content. Games included throughout the training were particularly popular with participants. The Menstrupedia comic was described as 'fun to read', with simple and understandable language.

The students expressed that if their schoolbooks also had similar pictures and stories, it would have been easier for them to learn and take interest in SRH issues. They felt that pictures, video, storytelling are better media to facilitate teaching SRH issues. The students shared that the lessons learnt in the Menstrupedia classes were more easily grasped than the routine lessons taught by the teachers because the Menstrupedia theme had a story and was taught in an interesting manner.

*"Our teacher only teaches the content inside the book- whereas in training, we learned so many extra things. Videos were shown in the training which made it easy to understand the content." (13 yo male)*

**Co-education during the training.** We provided the training to all the adolescents, irrespective of gender identity, together in the same session. Male participants expressed that they preferred co-educational SRH lessons because they felt it would help decrease stigma and help students to talk about SRH issues with the opposite gender.

*"If girls and boys aren't kept together for a class that teaches sexual and reproductive health, the girls will feel shy to express their problems at home too. So, boys and girls should be kept together." (12 yo male)*

However, the opposite view was expressed by some of the female participants who preferred separate SRH training sessions for girls and boys because they felt awkward asking sensitive questions in front of the boys. Some of the boys mentioned that although it is important to learn and discuss SRH issues, it was also equally important to share SRH issues only with the people whom they trust.

*"It is better to teach boys and girls together in a class, but boys laugh during the class and we feel awkward to ask questions . . ..so it is also good to separate boys and girls during SRH classes." (12yo, female)*

**Impact on MH practice and attitude.** The girls found the training useful and reported remembering and incorporating many of the instructions provided. The training not only provided advice regarding menstrual hygiene practices to the adolescents but also explained the reasons for following the instructions. This appeared to help the girls reflect critically on their own personal MH practices. A higher proportion of participants reported drying their washed cloths used for absorbing blood in sunlight after the training (70.5% post Menstrupedia intervention compared to 57.1% pre-intervention). Prior to the training, some participants were embarrassed to dry these cloths in the open air, but the training provided them with the information as to why it was necessary to do so, and thus they changed their practice. They also mentioned that they did not use soap internally in the vagina when taking showers after being informed of potential disruption to vaginal pH and microbial balance during the training session. They also mentioned that they found learning about ways of proper disposal of the pads

and about the exercises that could be done during their period useful. Participants also prepared themselves for menstruation beforehand by calculating the days of estimated date of menstruation as taught during the training.

> *"We didn't use to dry underwear and cloth used in menstruation under direct sunlight in past. We used to feel shy to do so. After reading the Menstrupedia comic book we learned to dry our cloths that are used during menstruation under direct sunlight." (13 yo female)*

Knowledge has an impact on practice and attitude. The girls were better informed about MH and thus were more confident and less shy about discussing MH issues. They shared that the training helped them to not to feel shy when talking about menstruation, to ask for pads if menstruation occurs in school, and to teach others what they have learnt. The girls mentioned that after the training, they have been able to overcome their shyness regarding talking about menstruation.

> *"We used to get shy buying Sanitary pads in the past. Now after reading the Menstrupedia comic we know that menstruation is a normal phenomenon and it's not the sin of god. Nowadays we don't feel shy to buy Sanitary pad. We realized that we need to aware people about this." (12 yo female)*

## Discussion

Almost all women and girls in Nepal live with restrictions to normal life and high levels of stigma surrounding menstruation [3]. Our study population was no exception, with 93.2% (n = 69/74) of girls reporting severe restrictive practices during menstruation, such as being forbidden to enter the kitchen, engage in normal communal social activities, touch male family members or even books. Such practices are deeply rooted traditions which have lifelong adverse consequences for socioeconomic and mental wellbeing. Changes in attitude and practice around menstruation can occur with educational interventions to increase knowledge, challenge taboos and facilitate discussion. The majority of adolescent girls obtain their first information surrounding menstruation from their mothers or other female relatives, and therefore increased knowledge, or persistent stigmas resulting from lack of education, will be passed to future generations.

We have demonstrated that a dynamic, interactive menstrual hygiene management lesson for adolescent students based on the Menstrupedia comic was effective in improving knowledge and attitude, and reducing persistent cultural stigmas surrounding menstruation. Knowledge of the physiological process and strategies for managing menstruation can foster a more positive attitude among women and girls [29]. The Menstrupedia-based training provided the students with the knowledge and facts about menstruation which reduced negative attitudes and perceptions around menstruation. Both knowledge and attitude scores increased significantly after the Menstrupedia-based training was delivered. This finding aligns with the results of other studies in Bangladesh and Indonesia [30, 31]. Our intervention included a one-time training session, the study from Bangladesh used a longer-term intervention with twelve lessons of 45 minutes each and the Indonesian study only provided students with a booklet for personal reading. Although the modality of the educational intervention varied in these studies, all of these studies showed improvement in menstrual knowledge and attitude following delivery of the interventions. A positive attitude towards menstruation can help women and girls have more positive experiences related to menstruation [32]. With a positive attitude towards menstruation, boys and men can help and support the females to manage

menstruation effectively in different settings including household, community, school, and work [33], and discontinue negative harmful behaviours towards menstruating women and girls.

Improved practices around managing mestrual cloths were reported, but surprisingly reporting of restrictive taboos increased in the post intervention evaluation. This may have been due to the female participants feeling more comfortable sharing menstrual taboos practiced in their home after the group discussions.

The audio visual and interactive components of the Menstrupedia training were particularly reported as effective by the students in the FGDs. Studies have shown that audiovisual media are effective tools for learning [34–37]. Students respond positively to audio visual medium which can effectively capture attention and increase interest of participants [38, 39]. The adolescents mentioned that they enjoyed watching the videos and wished that their teachers also used such media to teach, instead of the traditional method of only using textbooks. However, the majority of the schools included in the study did not have adequate equipment and/or audiovisual devices which prevented the teachers from using such methods. In the few schools which had audiovisual teaching facilities, students and teachers reported that few lessons were delivered using the resources. Frequent power outages, which also disrupted our training sessions, were a strong contributing factor to teachers not utilising the audio-visual aids. Reliability of the electricity supply in rural Nepal is increasing each year due to increased generation of hydropower and strengthening of the distribution network. Battery and generator backup could also be used to address these challenges during teaching delivery. The Menstrupedia comic is a physical book which can be distributed to students and used as the teaching resource for the lessons, but this approach is likely to be less engaging than the animated video, because the focus group discussion emphasized that the students particularly enjoyed the animated video format of the lessons.

Overall, levels of knowledge surrounding menstruation in our pre-intervention matched population were low. The majority of the participants did know know the cause of menstruation (63%, n = 269/427) and the interval of the menstrual cycle (73.7%, n = 315/427). Only 10.5% (n = 45/427) of the girls correctly responded that the menstrual blood originated from the uterus. Three quarters of the girls (75.8%; n = 44/64) obtained information about menstruation from their mothers. Over half of the girls (60.3%; n = 35/64) reported that their friends were a source of information regarding menstrual hygiene, while books (25.8%; n = 15/64) and teachers (27.6%; n = 16/64) were relatively minor sources of MH information. This is consistent with reports from other settings globally which report mothers, friends and teachers as the primary sources of information regarding menstruation [11, 40–43]. The information on menstruation provided by mothers, which shapes the attitudes of the children, is often based on cultural myths and may be incomplete which helps perpetuate negative and distorted perceptions and practices of menstruation [44].

The MH knowledge of adolescents in our study population was also significantly associated with literacy level of the father. It is important that the parents, especially the mothers, who are the primary source of information, have adequate and accurate information about menstruation to perpetuate improved MH and eliminate taboos, stigma and shame in the next generation. If the adults surrounding the adolescents do not change their attitude towards menstruation, they will reinforce the persistence of negative practices within the household and wider community. Thus, it is necessary to also provide educational interventions to parents, grandparents teachers and community influencers regarding MH. Unfortunately, due to limited resources we were not able to expand the scope of the intervention in this project to include the broader community. However, the Menstrupedia comic tool should also be evaluated as a resource for facilitating adult education on MH.

The MH knowledge of participants was also found to be significantly associated with the mother's occupation, although the the highest median score was among those with mothers in informal employment, possibly reflecting time available to spend interacting with children. However, the median attitude score was significantly lower in children of monthers who had informal employment. The reasons for this are unclear, but may be related to the social and cultural influences- as menstruation is affected by the social norms, expectations, and beliefs about how women should feel, act, and behave, during the menstrual cycle [45]. Ocupation is a marker of socioeconomic status and those with higher income jobs are likely to also have higher levels of information regarding topics such as menstruation which they are able to share with their daughters. An open discussion about sexual and reproductive health issues, including menstruation, is more likely to occur in educated families [46, 47].

Changing the deeply entrenched cultural stigmas surrounding menstruation requires long term, sustained approaches. Educational interventions such as the one used in this study act not only to provide information about menstruation and menstrual hygiene management but also as an advocacy program for MH. Class teachers were present in the classrooms when the training was delivered and developed their skills in regard to MH education. Teachers were able to see the high level of engagment by students during the interactive sessions. This could encourage the teachers to incorporate similar teaching formats to deliver future lessons. In addition, since friends were also found to be an important source of information, the students can benefit from a peer-to-peer education modality. Interventions such as the one applied in this research may facilitate more accurate sharing of peer-to-peer knowledge beyond the participants in the training. Evaluation of this was, however, beyond the funding scope of the present study, but will be important to include in future research evaluations of school-based MHM interventions.

MH education, even in the form of a short intervention, is effective in improving understanding and attitudes around menstruation among adolescents and facilitating the fundamental right to MH. Ideally, MH education is integrated in CSE within the school curriculum. However, even trained teachers are often initially embarrassed to deliver the curriculum content on SRH, exacerbated by the extreme cultural stigma surrounding menstruation in Nepali society. Thus, professional expert menstrual health trainers are more likely to be able to deliver an engaging teaching module on MH. Extensive efforts by NGOs to improve training of teachers to deliver MH and CSE modules within schools may help teachers deliver SRH education effectively. However, for the students to gain adequate MH information within the school, it is also imperative that schools provide a supportive environment in terms of including adolescent friendly SRH curriculums, providing relevant books or materials and most importantly training teachers adequately to have confidence in delivering high quality SRH education modules. Basic education on adolescent sexual and reproductive health has been included in the curriculum by the government of Nepal for all students from grades six through ten [48]. However, sex education is reported to be poorly implemented in most public schools in the country. Teachers face barriers such as lack of adequate teaching materials and lack of school and community support for teachers to provide proper SRH lessons in schools. These barriers result in poor quality of sex and reproductive health education [49]. Therefore, there is a need for teacher training programs on SRH education to boost teacher confidence and reduce information barriers.

The MH knowledge of participants were also found to be significantly associated with gender, with a higher median score for girls. Since girls go through menstruation, they have first hand experience and a more significant motivation to seek out information regarding menstruation. Boys generally obtain MHM information from informal sources such as overhearing conversations or observing cultural rituals. The weaknesses in the school-based SRH

curriculum around menstruation can therefore particularly exclude boys from understanding menstruation [50]. The informal means of information provision regarding menstruation can perpetuate negative attitudes and cultural myths which in turn precipitates period-based teasing of female peers [51]. Thus, it is vital to include boys and men in discussions of menstruation in both school and home environments, so that they are able to understand menstruation and be allies in eliminating period stigma for future generations.

Menstrual knowledge and attitude were also found to be significantly different between the groups of adolescents according to the religion followed. Participants from families practicing Buddhism scored lowest. Only 6.2% (n = 4/64) of the girls reported following no restrictions at home during mestruation. Religious perceptions and restrictions during menstruation strongly influence attitudes towards menstruation which may be the reason why there was a significant difference in the attitude amongst adolescents following different religions. As a direct consequence of the shame and taboos around menstruation, hygiene facilities are inadequate to allow dignity during menstruation in most public spaces of Nepal, including schools. This is reflected in the fact that over 18% of the girls in our study reported having to take menstrual materials home for disposal. This indicates that not all girls are comfortable disposing their menstrual materials in the school, which could be due to the lack of facilities for washing and drying if they are using cloths or because they are not comfortable disposing of their pads in the dustbins, which are often located in a communal area rather than privately within the cubicle. Poor menstrual hygiene practices can cause reproductive and urinary tract infections and can be a significant predictor of many gynecological problems [52]. Schools can help the girls be comfortable managing their MHM materials by providing proper water and sanitation provisions. Minor alterations such as relocating disposal bins within each cubicle can also significantly enhance MHM practices. This will both help the girls dispose of their MHM materials without embarasement as well as preventing clogging of the drainage system due to throwing of pads directly into latrines with inadequate plumbing.

Although majority of the girls hung their washed cloths used during menstruation out in the sun, 23.5% (n = 4/17) were wary of people seeing their menstrual material and thus hid it and many (n = 8/17;47.1%) covered the cloth with another layer of cloth when drying. This finding is similar to many studies conducted in the South Asian context, where women frequently report that they hang the menstrual cloths to dry with their clothes, but cover them with another item, so that they cannot be seen, or dry them in areas where no one will see the cloths [53, 54]. Girls and women often do this due to social restrictions, and taboos—which may lead to the reuse of material that has not been adequately sanitized. Bacterial vaginosis (BV) and reproductive tract infections (RTIs) have been reported to be more common in women with unhygienic menstrual hygiene management practices [55, 56]. These unhygenic practices can include improper drying of the cloths which are used as the absorbent material. Unsanitary practices during menstruation may be a result of lack of appropriate and sufficient information regarding menstrual hygiene. This may ultimately increase a woman's susceptibility to reproductive tract infections and longer term gynecological complications.

Students showed improvement in the understanding of menstruation, its cause, interval between menstrual cycles and the average age at menarche following the Menstrupedia training. However, the training did not increase the proportion of students responding correctly to two questions. The first was a question regarding the source of menstrual blood. This may have been due to poor wording of the question, as a consequence of our limited evaluation of the questionnaire prior to the study. Some students may have interpreted the question to be regarding the exit channel for the menstrual blood from the body. Low general knowledge of female anatomy may also be responsible for the lack of clarity in responses to this question. The second question which failed to show an increase in correct responses was regarding the

normal duration of menstruation. This may have been due to confusion between the various numbers given during the training for different aspects of the menstrual cycle, including the period between two menstrual cycles, age of menarche and normal duration of menstruation. It will be important to establish the reasons for these gaps in knowledge gain, and to refine the training to address the issues in future use of the Menstrupedia intervention.

Pretesting the questionnaire and the training could have improved the clarity of the questions and responses. However, we were limited in our ability to pretest both the Menstrupedia training and the evaluation questionnaire due to the COVID pandemic situation. However, each question and the response options were discussed with the students in detail during administration of the questionnaire. Some of the students may have given socially acceptable answers leading to response bias, despite being informed that the answers were not going to be shared with anyone in the school. This is suggested by the increased reporting of some menstrual stigma practices after the training, such as exclusion from the kitchen during menstruation. This may have been due either to an increased willingness to report stigmatizing practice within the family, or alternatively due to an increased recognition of such practices as stigmatizing. Two experienced expert trainers from Putali Nepal delivered the Menstrupedia training. The outcome of the intervention may have been dependent on the skills and ability of the trainers, and it will be necessary to evaluate variation in trainer outcomes during any scale-up of the intervention.

## Conclusion

This study has highlighted that an engaging SRH educational intervention, such as the Menstrupedia comic has a positive impact on knowledge and attitude towards menstruation. It is necessary that parents and teachers take an active approach to educate adolescents, as they have many questions regarding their sexual and reproductive health. Educated parents, can impart menstrual knowledge to their children, which in turn can lead to positive perceptions towards menstruation. The study also highlighted that lack of sanitation facilities was a major barrier to good MH practice in Nepali schools, with over half of girls reporting that no soap was available in toilets. It is vital for schools to provide proper water and sanitation facilities to the students so that they can maintain proper hygiene practices during their menstruation.

## Acknowledgments

The research team expresses gratitude to the all the participating schools' principals and teachers for their cooperation which was essential for the successful conduction of the training classes and data collection for the research. We would also like to thank Ms. Sujata Adhikari who helped during the data entry of the paper-based data into ODK software.

## Author Contributions

**Conceptualization:** Swastika Shrestha, Saki Thapa, Maxine Caws.

**Data curation:** Swastika Shrestha.

**Formal analysis:** Swastika Shrestha.

**Investigation:** Swastika Shrestha, Saki Thapa, Bikram Bucha, Safal Kunwar, Bigyan Subedi.

**Methodology:** Swastika Shrestha, Saki Thapa.

**Project administration:** Swastika Shrestha, Bikram Bucha.

**Resources:** Aishwarya Rani Singh, Durga Datta Chapagain, Raghu Dhital.

**Software:** Swastika Shrestha.

**Supervision:** Maxine Caws.

**Validation:** Swastika Shrestha, Saki Thapa.

**Visualization:** Swastika Shrestha, Maxine Caws.

**Writing – original draft:** Swastika Shrestha.

**Writing – review & editing:** Swastika Shrestha, Maxine Caws.

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
