## [Decision Letter · Decision Letter 0]

14 Aug 2024

PONE-D-24-27175Effectiveness of menstrual hygiene management training to enhance knowledge, attitude, and practice among adolescents in Sindhupalchowk, Nepal.PLOS ONE

Dear Dr. Shrestha,

Thank you for submitting your manuscript to PLOS ONE. After careful consideration, we feel that it has merit but does not fully meet PLOS ONE’s publication criteria as it currently stands. Therefore, we invite you to submit a revised version of the manuscript that addresses the points raised during the review process.

Both reviewers have provided a thorough set of comments which need to be carefully addressed before the paper can be accepted.

We look forward to receiving your revised manuscript.

Kind regards,

Alison Parker

Academic Editor

PLOS ONE

Journal Requirements:

3. Thank you for stating the following financial disclosure: The research was funded by a personal funding made to the Birat Nepal Medical Trust by Mr Frank Guthrie. 

5. We note you have included a table to which you do not refer in the text of your manuscript. Please ensure that you refer to Table 5 in your text; if accepted, production will need this reference to link the reader to the Table.

Reviewers' comments:

Reviewer's Responses to Questions

**Comments to the Author**

1. Is the manuscript technically sound, and do the data support the conclusions?

Reviewer #1: Partly

Reviewer #2: Yes

2. Has the statistical analysis been performed appropriately and rigorously? 

Reviewer #1: Yes

Reviewer #2: No

3. Have the authors made all data underlying the findings in their manuscript fully available?

Reviewer #1: Yes

Reviewer #2: Yes

4. Is the manuscript presented in an intelligible fashion and written in standard English?

Reviewer #1: Yes

Reviewer #2: Yes

5. Review Comments to the Author

Reviewer #1: This is an interesting paper about a MH intervention. The large sample size resulted in significant differences before and after however in reality the improvement was minimal by one point for knowledge ? the max was 6 and the baseline 2/3. the median post was 3 compared to 2. So they barely made 50%. I would have assumed the intervention would be very focused on the knowledge presented in the questions. In this case i would suggest if one particular question there was a marked improvement while others not? what was it that they did not grasp. Similarly the attitudes was significant in improvement but also when looking at the scores had not really risen that much when the max score could be 70. I would have expected this lack of real change to be discussed outside of the claims of significant differences.

It seemed with the cultural issues in play from the tables that really the parents and teachers needed the intervention more than the girls. The girls may have changed a little but its the system that needs to change to ensure teachers who are there daily can support and that parents ensure girls are not stigmatised during their menstrual cycle. There was no discussion about this and potential difficulties to change attitudes if parents and communities held views and opinions that treated girls so differently when menstruating.

There was a lot of small grammatical errors like full stops not at exact end of sentence on page 12.. On page 14 it says the majority of girls when the % was only 55.4 % which is rather just over half.

Table 4 title should end "who had reached menarche, not who had their periods. This table seems very long.

I was a bit confused about the discussion around Table 4 as this was baseline? or after intervention? did you look at any of these items after the intervention? it seems odd presented at the end.

Reviewer #2: Abstract

Please give the sample size, age group, who delivered the training, and a brief description of the knowledge and attitudes scores used. Also please briefly describe the statistical methods used.

Introduction

I suggest you use the phrase Menstrual Health throughout the paper, and cite the paper with the definition of this (Hennegan 2021)

Lines 54/55 – your sentence talks about menstruation “around the world” but your references are a systematic review from India and two individual studies. Why not cite a global systematic reviews of menstrual knowledge among adolescents in LMICs here e.g. the Chandra-Mouli 2017 paper, Coast 2019, Hennegan 2019 (currently your reference 5). Simlarly reference 7 seems a bit ‘random’ as a single study from Nigeria. Best to cite studies from Nepal or neighbouring countries, or systematic reviews.

Lines 64/65: Similarly for educational interventions for MHM you could cite Evans et al 2021.

Please be clearer about the evidence gap that you are filling with your study – what does it add to the existing literature?

Methods

Line 78: Why do you call this a cross-sectional study when it is a pre-post longitudinal study?

Lines 86: Please give a little more detail about the intervention e.g. how long was the video, was it shown in class? How many students watched? Did boys & girls watch together and discuss together? (you give this information later but it should be in the methods).

Lines 92-96: How did you select the schools? Did any refuse?

Lines 95: Were all students in the selected classes eligible, or did you select students in each class?

Lines 112: How did you assess whether the students correctly understood the meaning of the questions? Did you undertake any cognitive testing prior to the survey?

Lines 113: Why did you have a time limit of 45 minutes?

Lines 108: Did you consider using validated tools on menstrual health practices such as the self-efficacy tool validated in Bangladesh (Hunter et al 2021?). Or the Menstrual practice needs scale (Hennegan 2020)

Please include a sample size calculation

The quantitative analysis should adjust for within-school clustering (e.g. using mixed-effects linear regression). It’s not clear in the methods if the outcome was change in score, or endline score adjusted for baseline.

Why did you collect endline data from participants who were not seen at baseline?

Table 2 should give the mean score at baseline and endline for each exposure level, and the (adjusted) mean difference with a 95%CI. It is not sufficient to just give a p-value as this doesn’t tell us about the magnitude or direction of effect. The results text also needs more detail about scores. Why do you not show the results of the knowledge & attitudes in a Table, like you do for practices? You give more details in the discussion but these should be in the results.

Were there specific questions that were poorly answered initially and then improved? Please show the results more clearly e.g. a bar chart with the % answering correctly pre- and post.

It is not clear what the “correct” answer is in Table 5.

Discussion

This is good but often goes beyond the findings of the study e.g. you talk about WASH practices which improve MHM but your intervention did not include WASH improvements in school.

Given how deeply rooted the menstrual-related restrictions and attitudes are in Nepal, it seems surprising that a brief intervention can be effective – please comment on this and whether you expect it to be sustained. Also it would have been good to include FGDs among parents as I expect their knowledge & attitudes are very important?

Given the problems with audiovisual materials, how would you advise this kind of intervention is scaled up?

Line 378 – this sentence is not complete.

Please include discussion of whether MH education should be undertaken by teachers and integrated with puberty/SRH education, or by NGOs or others?

Please also include a paragraph on the limitations of the study, and compare the effects you found with those of other educational interventions.

6. PLOS authors have the option to publish the peer review history of their article (what does this mean?). If published, this will include your full peer review and any attached files.

Reviewer #1: No

Reviewer #2: No

---

## [Author Response · Author response to Decision Letter 0]

19 Sep 2024

Dear Editor,

Thank you for the opportunity to revise our manuscript and submit the updated version. We also thank the reviewers for their generous comments on the manuscript. We have now edited the manuscript to address their concerns and present the response to the journal and the reviewers’ comments below. We hope that the manuscript is now suitable for publication.

Yours sincerely,

Dr Swastika Shrestha

On behalf of all the authors

Journal’s comments:

Thank you for the suggestion. We have ensured that the manuscript meets PLOS ONE’s style requirements.

Apologies for the typological errors in the manuscript. It was likely due to multiple versions of the manuscript which were made in track change mode. The revised manuscript has been copyedited by Dr Maxine Caws, Senior Researcher, Liverpool School of Tropical Medicine, UK.

3. Thank you for stating the following financial disclosure: The research was funded by a personal funding made to the Birat Nepal Medical Trust by Mr Frank Guthrie. Please state what role the funders took in the study.

The research was kindly funded by Mr Frank Guthrie. We state that “The funders had no role in study design, data collection and analysis, decision to publish, or preparation of the manuscript." 

We are not able to provide public access to the full dataset due to the sensitive nature of the topic in Nepali culture. We did not request permission to share transcripts from participants, or from the Nepal Health Research Council (Nepal Government authority responsible for approving all studies involving human participants in Nepal). Indeed, the adolescent participants were assured of confidentiality before participating, with a statement that data would only be reported in aggregate. 

We state that "Data cannot be shared publicly because consent was not obtained for public data sharing of full transcripts during the informed consent process, due to the sensitive nature of the questions. Data is securely stored in controlled access secure data storage at Birat Nepal Medical Trust (BNMT). Data is available from the BNMT data access Committee (contact: Kritika Dixit, research manager BNMT; kritika@bnmt.org.np) for researchers who meet the criteria for access to confidential data.”

5. We note you have included a table to which you do not refer in the text of your manuscript. Please ensure that you refer to Table 5 in your text; if accepted, production will need this reference to link the reader to the Table.

Apologies for the error. We have now referred table 5 in the text of the manuscript in Page 13 line 230.

Reviewer #1 comments: 

1. This is an interesting paper about a MH intervention. The large sample size resulted in significant differences before and after however in reality the improvement was minimal by one point for knowledge ? the max was 6 and the baseline 2/3. the median post was 3 compared to 2. So they barely made 50%. I would have assumed the intervention would be very focused on the knowledge presented in the questions. In this case i would suggest if one particular question there was a marked improvement while others not? what was it that they did not grasp. Similarly the attitudes was significant in improvement but also when looking at the scores had not really risen that much when the max score could be 70. I would have expected this lack of real change to be discussed outside of the claims of significant differences.

Thank you for the important observation and the comment. The one-point increase in median score is a significant improvement, given the grading of only 1-6 in scores. The increased knowledge score was due to improved understanding of the cause of menstruation, the duration of the cycle and average age at menarche among the students. The knowledge of the students did not improve regarding the source of menstrual blood and the normal duration of bleeding. These were important aspects of the training which need revision to improve clarity and knowledge retention. We were limited in our ability to pretest both the Menstrupedia training and the evaluation questionnaire due to the COVID pandemic situation. We agree that these are important and valuable observations to facilitate refinement of the training material, and we have incorporated this into the discussion text in lines 466-477 as follows:

“Students showed improvement in the understanding of menstruation, its cause, interval between menstrual cycles and the average age at menarche following the Menstrupedia training. However, the training did not increase the proportion of students responding correctly to two questions. The first was a question regarding the source of menstrual blood. This may have been due to poor wording of the question, as a consequence of our limited evaluation of the questionnaire prior to the study. Some students may have interpreted the question to be regarding the exit channel for the menstrual blood from the body. Low general knowledge of female anatomy may also be responsible for the lack of clarity in responses to this question. The second question which failed to show an increase in correct responses was regarding the normal duration of menstruation. This may have been due to confusion between the various numbers given during the training for different aspects of the menstrual cycle, including the period between two menstrual cycles, age of menarche and normal duration of menstruation. It will be important to establish the reasons for these gaps in knowledge gain, and to refine the training to address the issues in future use of the Menstrupedia intervention.”

2. It seemed with the cultural issues in play from the tables that really the parents and teachers needed the intervention more than the girls. The girls may have changed a little but its the system that needs to change to ensure teachers who are there daily can support and that parents ensure girls are not stigmatised during their menstrual cycle. There was no discussion about this and potential difficulties to change attitudes if parents and communities held views and opinions that treated girls so differently when menstruating.

Thank you for the comment. We agree that this is a broad societal issue that requires a change in knowledge, attitude and practice across all sectors of society to eliminate harmful stigma and resulting practice. Due to limited funding available we were only able to test the Menstrupedia intervention as a school training module at this time. We are actively seeking funding for broader interventions on menstrual health and many NGOs are working in this space to address the issues across South Asian and African societies with such taboos. Indeed, it is often grandparents and priests who reinforce the persistence of such practices. We have elaborated on this topic in the discussion section from line number 388-393 as follows:

“If the adults surrounding the adolescents do not change their attitude towards menstruation, they will reinforce the persistence of negative practices within the household and wider community. Thus, it is necessary to also provide educational interventions to parents, grandparents teachers and community influencers regarding MH. Unfortunately, due to limited resources we were not able to expand the scope of the intervention in this project to include the broader community. However, the Menstrupedia comic tool should also be evaluated as a resource for facilitating adult education on MH.”

3. There was a lot of small grammatical errors like full stops not at exact end of sentence on page 12.

We apologize for these errors and have corrected the errors throughout the manuscript.

4. On page 14 it says the majority of girls when the % was only 55.4 % which is rather just over half.

Thank you for the comment. We have revised this text to incorporate the suggested change, and corrected the % value, in page 15-line number 236 as follows:

“Over half of the girls (n=37/64; 57.8%) reported that their schools had the provision of only water and no soap available for handwashing.”

5. Table 4 title should end "who had reached menarche, not who had their periods. This table seems very long. 

Thank you for the comment. We agree and have revised the table title as suggested (line 241). We have simplified the data in table 4 and presented the breakdown of data for menstrual practice both pre and post intervention in table 5 to enable the reader to understand the menstrual health practice, and the areas which are challenging in the Nepali context. We believe this is important information arising from the study, but we are happy to transfer this to supplementary data if preferred by the editor. Final formatting of the table in print will reduce the apparent length. 

6. I was a bit confused about the discussion around Table 4 as this was baseline? or after intervention? did you look at any of these items after the intervention? it seems odd presented at the end.

Thank you for the comment and apologies for the confusion. Table 4 is a post-intervention table. The Pre and posttest comparison table has been presented in table 5. However, no statistical test has been performed to compare these scores as some of the practices are the result of limitations in available resources/facilities, which are not necessarily incorrect in the context in which the girls are living. Thus, we have not scored the practice as correct or incorrect. 

Reviewer #2 comments: 

Abstract

1. Please give the sample size, age group, who delivered the training, and a brief description of the knowledge and attitudes scores used. Also please briefly describe the statistical methods used.

Thank you for the comment. We have added the suggested details to the abstract now in line number 23-32 as follows:

“Methods: A school-based study was conducted in Indrawati rural municipality of Sindhupalchowk district in Nepal. 427 participants (175 boys and 252 girls), aged 11-13, completed a questionnaire evaluating MH KAP before receiving a structured training module on MH provided by experienced trainers from Putali Nepal using the Menstrupedia tool. The questionnaire was repeated one month after the training. Pre and post intervention scores were compared to determine the effect of the intervention. Focus group discussions were also conducted to understand the perceptions of participants toward SRH teaching. 

Association of independent socio-demographic with dependent variables- knowledge and attitude -towards menstrual health were analyzed using MANOVA test. The Wilcoxon signed-rank test was used to compare the median outcome of the pre and post-test attitude and knowledge. The maximum possible score was 6 for MH knowledge. The total attitude score ranged 14 to 70.”

Introduction

2. I suggest you use the phrase Menstrual Health throughout the paper, and cite the paper with the definition of this (Hennegan 2021)

Thank you for the comment. We have edited the text to Menstrual Health and added the citation for the definition (lines 48-50) as follows:

“This inevitably affects the MH of women and girls- which is defined as a state of complete physical, mental, and social well-being and not merely the absence of disease or infirmity, in relation to the menstrual cycle.”

3. Lines 54/55 – your sentence talks about menstruation “around the world” but your references are a systematic review from India and two individual studies. Why not cite a global systematic reviews of menstrual knowledge among adolescents in LMICs here e.g. the Chandra-Mouli 2017 paper, Coast 2019, Hennegan 2019 (currently your reference 5). Simlarly reference 7 seems a bit ‘random’ as a single study from Nigeria. Best to cite studies from Nepal or neighbouring countries, or systematic reviews.

Thank you for the suggestion. We have added the suggested references (line 44).

4. Lines 64/65: Similarly for educational interventions for MHM you could cite Evans et al 2021.

Thank you for the suggestion. We have added Evans et al 2021 as a reference in line 55.

5. Please be clearer about the evidence gap that you are filling with your study – what does it add to the existing literature?

We aimed to evaluate the Menstrupedia tool as a school-based intervention to improve knowledge surrounding menstrual health in school-based adolescents in rural Nepal. We faced significant challenges in the planned implementation due to the COVID pandemic situation, which affected the rigor of the study. In particular, school attendance was significantly disrupted, resulting in disparity between the children attending during pre- and post- intervention evaluation. However, despite this, we were able to show that the Menstrupedia training was a popular and engaging format with students, and the relatively short training significantly increased knowledge and attitude scores. The intervention warrants a larger scale comprehensive impact evaluation. There are few studies showing effectiveness of interventions for menstrual health knowledge among adolescents in south Asia, while there are many studies of stigma and impact on girls’ school attendance. We believe the development of evidence-based interventions and the generation of evidence supporting implementation is crucial for supporting political commitment and investment in changing school-based teaching for menstrual health and the broader comprehensive sexuality education curriculum. We have added clarification of the study purpose to the text (lines 66-75) as follows:

“Underpinning Sustainable Development Goals (SDG) (18), MH is a right for every woman and girl. Imparting MH education is an essential step towards ensuring the right to MH for women and girls. Although a part of the school curriculum, schools in Nepal have not been able to deliver the SRH education effectively. In such context, development and evaluation of evidence-based interventions is crucial for supporting political commitment and investment in improving school-based teaching for MH and the broader comprehensive sexuality education (CSE) curriculum. Despite the severe and deeply rooted stigma surrounding menstruation in Nepal, which has multidimensional negative impacts on the lifelong wellbeing of women and girls, there is a paucity of published studies evaluating MH educational interventions in the Nepalese context. Therefore, we designed the present study to evaluate the effectiveness of a school-based menstrual education program in improving menstrual knowledge and attitude among adolescents in Nepal.”

Methods

6. Line 78: Why do you call this a cross-sectional study when it is a pre-post longitudinal study?

Thank you for the comment. We have edited the text to ‘pre-post longitudinal study’ in place of ‘cross-sectional study’ (line 78) as follows:

“This is a pre-post longitudinal school-based study conducted in Indrawati rural municipality of Sindhupalchowk district, Nepal where there is a high prevalence of underage marriage.” 

7. Lines 86: Please give a little more detail about the intervention e.g. how long was the video, was it shown in class? How many students watched? Did boys & girls watch together and discuss together? (you give this information later but it should be in the methods).

Thank you for the suggestion, we have added further details of the intervention to the methods text (lines 86-89) as follows:

 “The adolescents were provided with inf

---

## [Editor Report · Decision Letter 1]

24 Oct 2024

Effectiveness of menstrual hygiene management training to enhance knowledge, attitude, and practice among adolescents in Sindhupalchowk, Nepal.

PONE-D-24-27175R1

Dear Dr. Shrestha,

We’re pleased to inform you that your manuscript has been judged scientifically suitable for publication and will be formally accepted for publication once it meets all outstanding technical requirements.

Kind regards,

Alison Parker

Academic Editor

PLOS ONE
---

## [Editor Report · Acceptance letter]

31 Oct 2024

PONE-D-24-27175R1 

PLOS ONE

Dear Dr. Shrestha, 

I'm pleased to inform you that your manuscript has been deemed suitable for publication in PLOS ONE. Congratulations! Your manuscript is now being handed over to our production team.

Kind regards, 

on behalf of

Dr. Alison Parker 

Academic Editor

PLOS ONE